# Why Does the Automation Say One Thing but Does Something Else? Effect of the Feedback Consistency and the Timing of Error on Trust in Automated Driving

J. B. Manchon [1,2,*], Romane Beaufort [3], Mercedes Bueno [1] and Jordan Navarro [2,4]

1    VEDECOM Institute, 78000 Versailles, France
2    Laboratoire d'Etude des Mécanismes Cognitifs (EA 3082), University Lyon 2, 69007 Lyon, France
3    Institut de Démographie de l'Université Paris 1 (IDUP), 75013 Paris, France
4    Institut Universitaire de France, 75005 Paris, France
*    Correspondence: jb.manchon@univ-lyon2.fr

**Abstract:** Driving automation deeply modifies the role of the human operator behind the steering wheel. Trust is required for drivers to engage in such automation, and this trust also seems to be a determinant of drivers' behaviors during automated drives. On the one hand, first experiences with automation, either positive or not, are essential for drivers to calibrate their level of trust. On the other hand, an automation that provides feedback about its own level of capability to handle a specific driving situation may also help drivers to calibrate their level of trust. The reported experiment was undertaken to examine how the combination of these two effects will impact the driver trust calibration process. Four groups of drivers were randomly created. Each experienced either an early (i.e., directly after the beginning of the drive) or a late (i.e., directly before the end of it) critical situation that was poorly handled by the automation. In addition, they experienced either a consistent continuous feedback (i.e., that always correctly informed them about the situation), or an inconsistent one (i.e., that sometimes indicated dangers when there were none) during an automated drive in a driving simulator. Results showed the early- and poorly-handled critical situation had an enduring negative effect on drivers' trust development compared to drivers who did not experience it. While being correctly understood, inconsistent feedback did not have an effect on trust during properly managed situations. These results suggest that the performance of the automation has the most severe influence on trust, and the automation's feedback does not necessarily have the ability to influence drivers' trust calibration during automated driving.

**Keywords:** trust in automation; automated driving; feedback; driver's behavior; critical situations





## 1. Introduction

Driving automation is ongoing and the higher levels of automated driving (level 4 and 5, [1]) should allow drivers to reduce the cognitive load related to driving and liberate time to engage in other activities. However, these benefits imply that drivers would be willing to delegate the driving task to a Highly Automated Driving (HAD) system that will be able to perform the dynamic driving task instead of them in specific environments, corresponding to level 4 [1]. Going from a fully manual task to an increasingly more automated one requires the drivers to trust automation [2]. This specific type of trust (i.e., Trust in Automation, TiA) has been shown to strongly influence and predict automation use and performance [3–6]. TiA is a central concept when one considers human–automation interaction from the human perspective. Indeed, TiA is a psychological construct that encapsulates human attitudes toward automation and directly influences reliance on automation.

Theoretical models have been developed to better understand TiA. A previous literature review exploring suitable models [7] led to the selection of one factor-centered model, the structure of which served as a basis to study trust-influencing factors. Hoff and

Bashir [8] proposed a model where trust is decomposed into three layers and described the factors influencing each of these trust layers. The first layer, dispositional trust, is linked to the human operator and evolves slowly during one's lifetime. The second layer, situational trust, is mainly linked to the context of interaction between the individual and the automation. It changes from one situation to another. The last layer, learned trust, is composed of two parts. The initial learned trust refers to the previous knowledge and expectations the operator has about the considered automation. It changes after each interaction when the new experience is processed by the operator. Finally, the dynamic learned trust depends on the automation's performance and features, and it evolves dynamically during the interaction. In the driving context, Trust in Automated Driving (TiAD) strongly relies on TiA and seems to play a major role in HAD acceptance, adoption [9,10], and proper use [11]. It therefore appears crucial to study the implications of this TiAD and the aspects that influence its dynamic calibration during HAD use. Here, the focus was set on dynamic learned trust to investigate how TiAD evolves when drivers actually interact with the HAD system. Previous experiments showed drivers' TiAD was negatively influenced by critical situations (i.e., a driving situation that requires the automation to perform an urgent maneuver in order to preserve passengers' safety) occurring during drives [12–14]. Hoff and Bashir [8] reported the timing of error, which is related to the automation performance, had a major influence on the operator's trust development. Critical situations that occur early during the interaction with automation have a greater negative influence than errors occurring after a positive experience with the automation [15,16]. In the driving context, errors may encompass a variety of automation malfunctions, such as unexpected Take-Over Requests (TORs) that decrease TiAD [12] or poorly managed situations [13].

In addition, studies have shown drivers sometimes had trouble to understand why the system managed the situations poorly [13]. Hoff and Bashir [8] also reported the transparency of the feedback provided by the automation was an influential factor related to its design features. Transparent feedback has an indirect positive influence on TiA, while less transparent feedback may confuse operators.

The transparency of the feedback can be described as the ability of individuals to correctly understand and evaluate the inner processes that lead an automation to behave in a certain manner [17]. Feedback that allows drivers to better estimate the HAD performance and malfunctions seems to promote a well-calibrated TiAD [12]. Moreover, continuous feedback indicating HAD uncertainty in various situations (e.g., in foggy weather) also supports TiAD calibration [18,19]. Understanding the ongoing performance of HAD therefore seems crucial for drivers to have trust in it.

Moreover, drivers' initial learned level of trust seemed to play a role in their further perception of critical situations during drives. Trustful drivers trusted the HAD system more during similar critical situations, compared to distrustful drivers [13,14]. In these previous experiments, drivers were recruited depending on their either high or low initial learned level of TiAD. This methodology allowed us to obtain two homogeneous groups and display strong results on drivers' TiAD evolution patterns, depending on situations encountered under HAD. Distrustful participants' level of trust increased more during the first minutes of interaction, compared with trustful participants. Moreover, distrustful participants' level of trust varied with a greater amplitude depending on the encountered critical situations. Participants' level of trust also impacted their visual behaviors during drives: people with a higher level of trust monitored the driving environment less, while people with a lower level of trust monitored it more.

The present experiment aims to evaluate both the influence of perceived understanding on TiAD and the dynamic evolution of TiAD depending on the timing of a critical situation poorly managed by the HAD system. These two effects are examined depending on drivers' initial learned level of trust, in a driving simulator, for the driver's very first experience with such automation.

The timing of the critical situation is expected to influence the dynamics of drivers' trust evolution. Drivers who experience an early critical situation are expected to have

a lower level of trust in the HAD system, and a slower trust increase during the rest of the drive, with more gazes directed toward the driving environment (H1). Inconsistent feedback is expected to slow the TiAD increase and the perceived understanding of the HAD system and increase the number of gazes directed toward the driving environment and the automation (H2). Finally, drivers who have a low initial learned level of TiAD are expected to have a lower level of trust during the drive (H3). Low initial learned level of TiAD was also expected to have more impact with an early critical situation (H4) or an inconsistent feedback (H5) compared with drivers who have a high initial learned level of TiAD.

## 2. Materials and Methods

### 2.1. Participants

Sixty-one drivers participated in this experiment (29 females, M = 41.97 years old, SD = 11.25, min = 21, max = 64). They all had normal or corrected-to-normal vision and audition, to ensure they would correctly perceive the automation feedback (because of the feedback nature, people with color vision deficiency were not included in this study). Participants had all carried a valid driver's license for a minimum of three years (M = 22.3, SD = 11.4, min = 3, max = 46). They reported that they drove regularly: 37 (60.7%) drove every day, 18 (29.5) at least once a week, five (8.2%) at least once a month, and one (1.6%) drove less frequently. Drivers' self-evaluated level of experience with advanced automated driving assistance systems was inquired. Eleven (18.0%) declared they had advanced knowledge of AD, 20 (32.8%) had intermediate knowledge, 21 (34.4%) basic knowledge, and nine (14.8%) declared they had no knowledge at all regarding HAD. Drivers also completed a trust scale form [13,14] to measure their initial learned level of TiAD.

### 2.2. Apparatus

The study was conducted in a static driving simulator equipped with three 2520 × 1440 mm panels, providing a 200° horizontal field of view. Three 7″ 16:9 LCD screens displayed a rear view of the driving environment, in place of real car mirrors. A 10″ 16:9 LCD screen was used to display the dashboard. A 10.1″ touchscreen was placed to the right of the steering wheel as the Human–Machine Interface (HMI). On the left sideband of the HMI, different pictograms were displayed to indicate the state of the HAD system and any relevant vehicle maneuvers, duplicating the information from the dashboard. The rest of the HMI displayed a tablet with an Android™ emulator with video games and Internet access (Figure 1). A colored RGB LED strip was positioned all around the edges of the cockpit and around the HMI to ensure the colored information would always be present in the drivers' field of view. The color of the LED strip varied depending on the state of the HAD system (Figure 2). The RGB LED strip was 4.1 m long and was composed of approximately 60 LEDs per meter, at an intensity of four to six lumens. The used RGB colors were white (RGB (255, 255, 255) for manual driving), blue (RGB (0, 0, 255) for available HAD), green (RGB (0, 255, 0) for engaged HAD in normal conditions), orange (RGB (255, 150, 0) when the HAD detected a potential problem, but was able to manage it), and red (RGB (255, 0, 0) for TORs). The SCANeR™ Studio 1.8 software (AV Simulation, France, https://www.avsimulation.fr, accessed on 6 July 2022), was used to run the driving simulation. The eye-tracking system was composed of four 60 Hz cameras recording data through the Smart-Eye Pro 6.2 software.

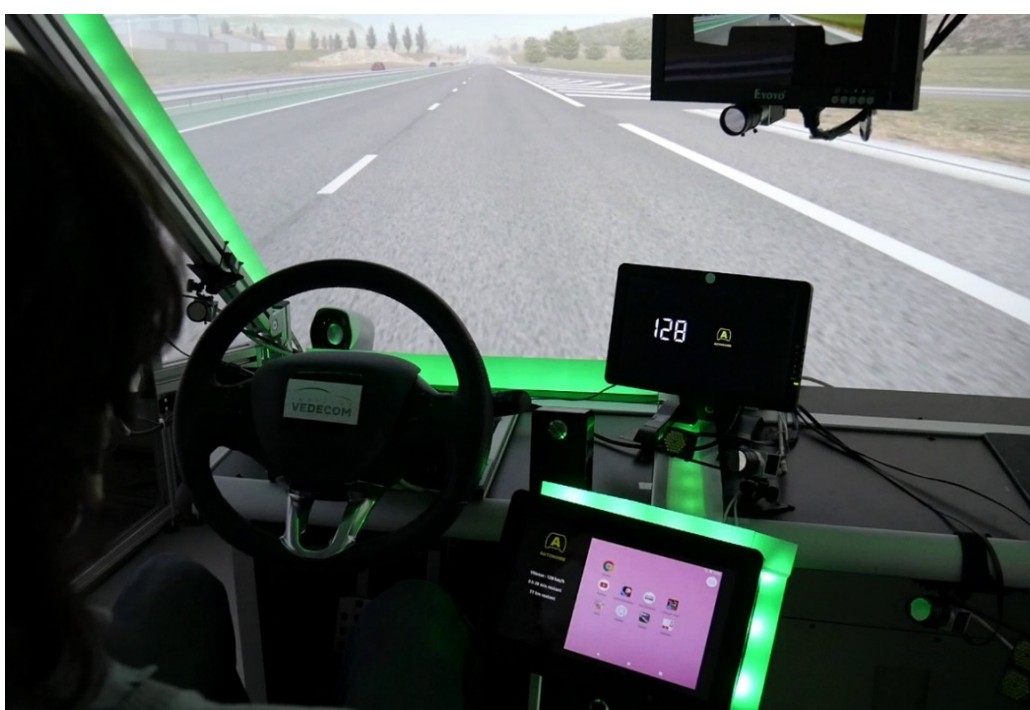

**Figure 1.** Close-up view of the apparatus (here, the LED stripe is green).

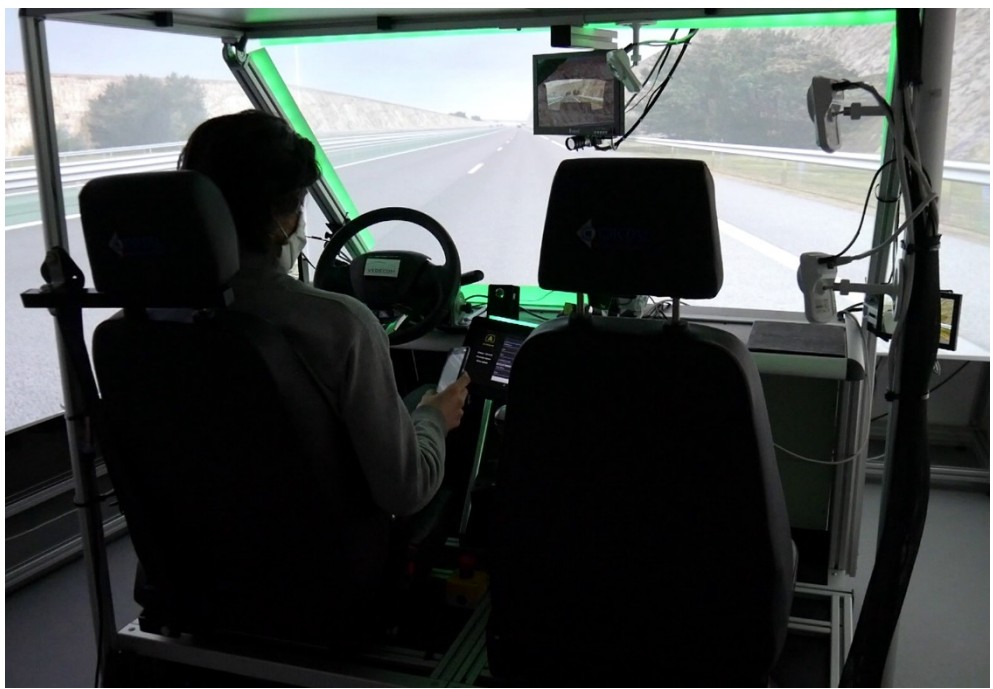

**Figure 2.** Wide view of the apparatus (the LED stripe is green).

*2.3. Procedure*

Participants arrived at the VEDECOM Institute, signed a consent form, and completed a trust scale form [13] to evaluate their initial learned level of trust (Table 1, Qi in Figure 3). The driving simulator was explained to ensure all participants correctly understood what each screen was displaying. Instructions were then given to participants. They were told they would experience a 35 min highly automated drive on the highway. The pictograms and colors indicating the several HAD states were explained, then a five-minute training was completed to allow drivers to familiarize themselves with the simulator and the HMI.

Participants were encouraged to behave in the same way they would in a real vehicle. They were not required to do any particular task, and they were free to engage in any non-driving-related activity they felt appropriate.

**Table 1.** Initial learned trust scale form [13]. * Answer was inverted for scoring.

|  | **Items** |
|---|---|
| 1 | I would feel safe in an automated vehicle. |
| 2 | The automated driving system provides me with more safety compared to manual driving. |
| 3 * | I would rather keep manual control of my vehicle than delegate it to the automated driving system on every occasion. |
| 4 | I would trust the automated driving system decisions. |
| 5 | I would trust the automated driving system capacities to manage complex driving situations. |
| 6 | If the weather conditions were bad (e.g., fog, glare, rain), I would delegate the driving task to the automated driving system. |
| 7 | Rather than monitoring the driving environment, I could focus on other activities confidently. |
| 8 | If driving was boring for me, I would rather delegate it to the automated driving system than do it myself. |
| 9 | I would delegate the driving to the automated driving system if I was tired. |

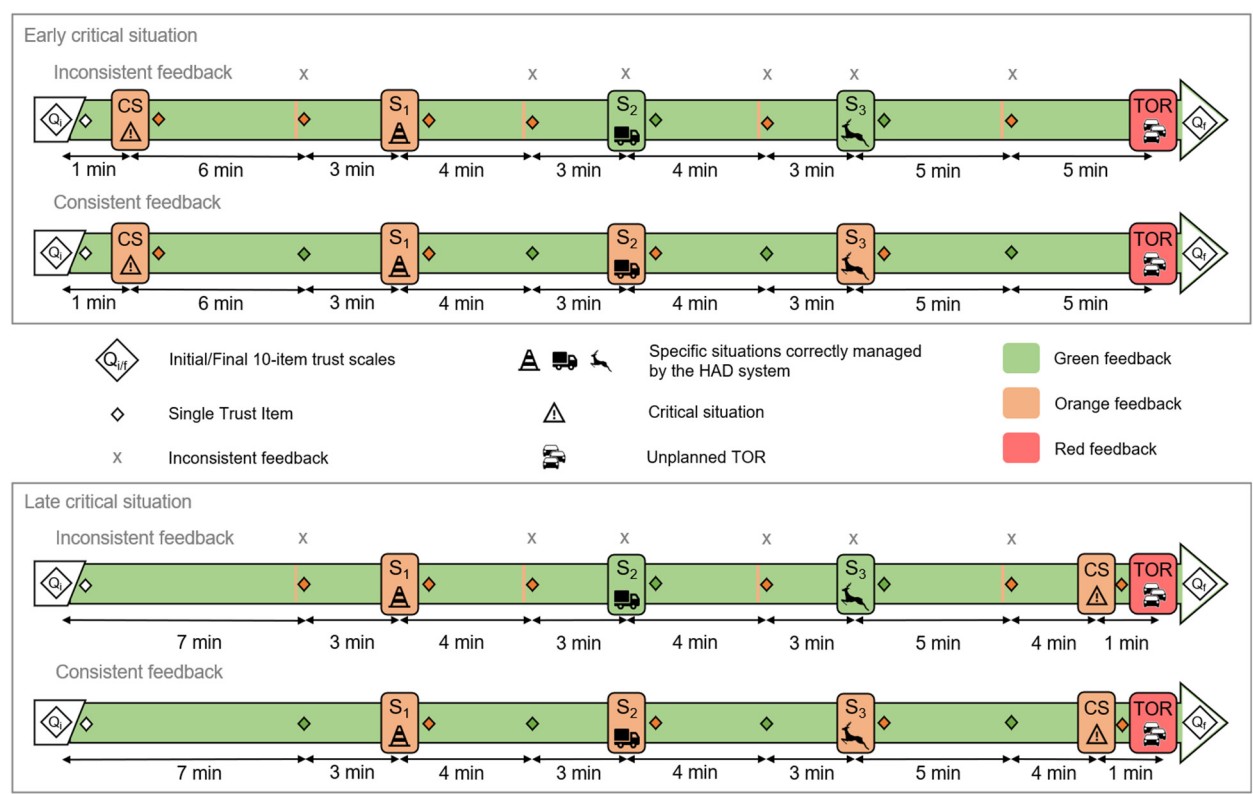

**Figure 3.** Procedure depending on the assigned conditions.

Participants were then randomly assigned to one of the four combinations of the two between-subject variables: the timing of the critical situation (early vs. late) and the feedback consistency (consistent vs. inconsistent), which will be detailed later. They were not aware of this.

Participants then started the experiment. They answered a first single trust item that served as a baseline after their training experience. One minute after the beginning of the drive, drivers who were assigned to the early critical situation condition experienced a missing lane marking situation for 350 m. This event was poorly managed by the vehicle

which zigzagged on the road with a maximum lateral lane shift of 1.2 m, until lane markings were found again (see Video S1: Critical situation (zigzag)). Drivers who were assigned to the other condition experienced a normal, monotonous highway driving during this time. After that, all drivers experienced three correctly managed situations, each spaced by several minutes of monotonous highway driving. First, a roadworks area forced the ego vehicle to decelerate (from 128 km/h to 90 km/h in five seconds) and change lanes to avoid this area (see Video S2: Critical situation (roadwork area)). Then, a slow truck forced the ego vehicle to decelerate (from 128 km/h to 80 km/h in four seconds) because the left lane was congested, until the truck had exited the highway. Finally, a deer crossing the road forced the ego vehicle to decelerate (from 128 km/h to 80 km/h in five seconds). Each of these situations was replicated from a previous experience that gave insights regarding situations that have a positive or a negative effect on drivers' level of TiAD [14]. Nine minutes after the crossing deer situation, drivers who were assigned to the late critical situation condition experienced the portion of the road with no lane marking, while the other drivers had a monotonous drive time. Last, all drivers experienced an unplanned TOR, indicated by a red color of the LEDs, an urgent sound, and a relevant pictogram.

During the drive, participants assigned to the inconsistent feedback condition experienced four false alerts during monotonous driving and two missing events during the last two correctly managed situations (the slow truck and the crossing deer situations). During the false alerts, the LED became orange, and a sound and a pictogram indicated that the automation had detected a situation on the road while there was nothing particular at these points. During the missing events, the feedback failed to detect the situation (i.e., green LED color instead of orange). All these situations were correctly detected for the consistent feedback group.

After each situation (both critical situations, CS in Figure 3, and correctly managed situations, S1, S2, and S3 in Figure 3) and after each moment of feedback conditions (both consistent and inconsistent; see ◊ in Figure 3), a single trust item inquiring drivers' current level of TiAD was displayed on the HMI (i.e., "How much do you trust the automated driving system?"). Another item inquiring drivers' perceived understanding of the HAD system behavior was also displayed (i.e., "How much did you understand the automated driving system reactions"). After the end of the drive, participants complated a 10-item trust scale form (the same that was used to assess their initial learned level of TiAD) to evaluate their final level of TiAD. They were finally debriefed, thanked, and rewarded.

### 2.4. Data Analysis

Dependent variables included subjective trust, perceived understanding of the HAD system, and visual strategies.

Single trust items and perceived understanding items, both ranging from zero to 100 were used. The 10-item trust scale was scored according to [14], and participants were divided into two groups (i.e., referred as Trustful and Distrustful as compared to the rest of the sample) based on a median-split.

Visual strategies data, or time spent looking towards specific Areas of Interest (AOI) were processed in percentages according to ISO 15007 recommendations (International Organization for Standardization, 2020). AOI were defined as "Road", "Rear Mirrors", "Landscape", "Dashboard", "HMI", "Android tablet", "Phone", and "Other" (for all glances directed somewhere else than the previously established AOI). "Road" and "Rear Mirrors" were grouped in the category "Road", "Dashboard" and "HMI" were grouped in the category "Dashboard", and "Android tablet" and "Phone" were grouped in the category "Non-Driving-Related Activities" for analysis.

Data were processed using R [20] and ggplot2 [21]. ANOVAs were run using the Greenhouse–Geisser correction.

*2.5. Design*

The following analyses were performed using a mixed experimental design. Pre-examination-confirmed data followed a normal distribution, without outliers. Between-subject variables were the initial learned level of trust: Trustful vs. Distrustful, the feedback consistency: Consistent vs. Inconsistent, and the timing of the critical situation: Early vs. Late. Within-subject variables were the color of the feedback: Green, Orange, or Red, and Time, which took different values depending on the analyses. In the ANOVAs analyses, Time was ten measurement points. In the linear models, Time was each passing minute from the start to the end of the experiment.

**3. Results**

*3.1. Trust and Perceived Understanding Questionnaires*

The median-split on the drivers' initial learned level of trust score resulted in two groups, which were subdivided into eight groups depending on the randomly assigned independent variables (i.e., feedback consistency and timing of error). Groups specificities are given in Table 2. Table 3 presents Bartlett's *t*-test that shows no evidence to claim that the groups' variances are not equal.

**Table 2.** Participants' initial and final level of trust. Q1 and Q3 are the first and third quartiles.

| Initial Level of Learned Trust | Feedback Consistency | Timing of Error | Trust Measurement | N | Age (Mean) | Trust | | | | | |
|---|---|---|---|---|---|---|---|---|---|---|---|
| | | | | | | Mean | Min. | Q1 | Median | Q3 | Max. |
| Trustful | Consistent | Early | Initial | 8 | 45.2 | 69.8 | 60 | 67.5 | 69 | 71 | 82 |
| | | | Final | | | 81.2 | 50 | 75.5 | 82 | 91.5 | 100 |
| Trustful | Consistent | Late | Initial | 8 | 41.1 | 75 | 58 | 71.5 | 73 | 80 | 90 |
| | | | Final | | | 94 | 72 | 80 | 87 | 90.5 | 100 |
| Trustful | Inconsistent | Early | Initial | 8 | 36.5 | 74.5 | 58 | 68.5 | 74 | 82.5 | 90 |
| | | | Final | | | 80.2 | 66 | 71 | 83 | 87 | 94 |
| Trustful | Inconsistent | Late | Initial | 9 | 43.3 | 72.9 | 58 | 66 | 70 | 82 | 90 |
| | | | Final | | | 80 | 54 | 68 | 82 | 86 | 100 |
| Distrustful | Consistent | Early | Initial | 7 | 40.7 | 41.4 | 10 | 42 | 44 | 50 | 52 |
| | | | Final | | | 52 | 30 | 34 | 54 | 68 | 76 |
| Distrustful | Consistent | Late | Initial | 7 | 42.6 | 45.1 | 30 | 45 | 46 | 49 | 52 |
| | | | Final | | | 69.7 | 52 | 61 | 64 | 75 | 100 |
| Distrustful | Inconsistent | Early | Initial | 7 | 41.6 | 47.1 | 36 | 41 | 46 | 55 | 56 |
| | | | Final | | | 65.1 | 46 | 49 | 58 | 80 | 94 |
| Distrustful | Inconsistent | Late | Initial | 7 | 44.7 | 39.1 | 24 | 31 | 38 | 47 | 56 |
| | | | Final | | | 69.4 | 38 | 57 | 72 | 74 | 100 |

**Table 3.** Bartlett's *t*-tests for participants' initial level of trust.

| Feedback Consistency | Timing of Error | Initial Level of Learned Trust | Bartlett's df | Bartlett's *t* | Bartlett's *p* |
|---|---|---|---|---|---|
| Consistent | Early | Trustful | 1 | 3.79 | 0.052 |
| | | Distrustful | | | |
| Consistent | Late | Trustful | 1 | 0.595 | 0.441 |
| | | Distrustful | | | |
| Inconsistent | Early | Trustful | 1 | 0.339 | 0.560 |
| | | Distrustful | | | |
| Inconsistent | Late | Trustful | 1 | 0.002 | 0.967 |
| | | Distrustful | | | |

A repeated measure ANOVA on single trust items showed the initial learned level of trust had an impact on the further level of trust during the drive, $F(1, 59) = 13.5$, $p < 0.001$,

$\eta_p^2 = 0.187$. Trustful drivers declared a higher trust in the automated driving system during the whole drive. Another repeated measure ANOVA was then run on the feedback consistency and the timing of the critical situation (Figure 4). The results showed a main effect of the Time, $F(4.86, 273.7) = 21.3$, $p < 0.001$, $\eta_p^2 = 0.272$, and an interaction between the Time and the timing of the critical situation, $F(4.86, 273.7) = 2.27$, $p < 0.05$, $\eta_p^2 = 0.038$. No other effects were significant. This result indicates drivers' trust varied during the drive, and this variation was different depending on whether drivers experienced an early or a late critical situation. Trust increased during the drive among drivers who experienced a late critical situation compared with drivers who experienced an early one. Trust dropped during the critical situation (regardless of if it was early or late) and during the TOR, compared to the other trust measurements.

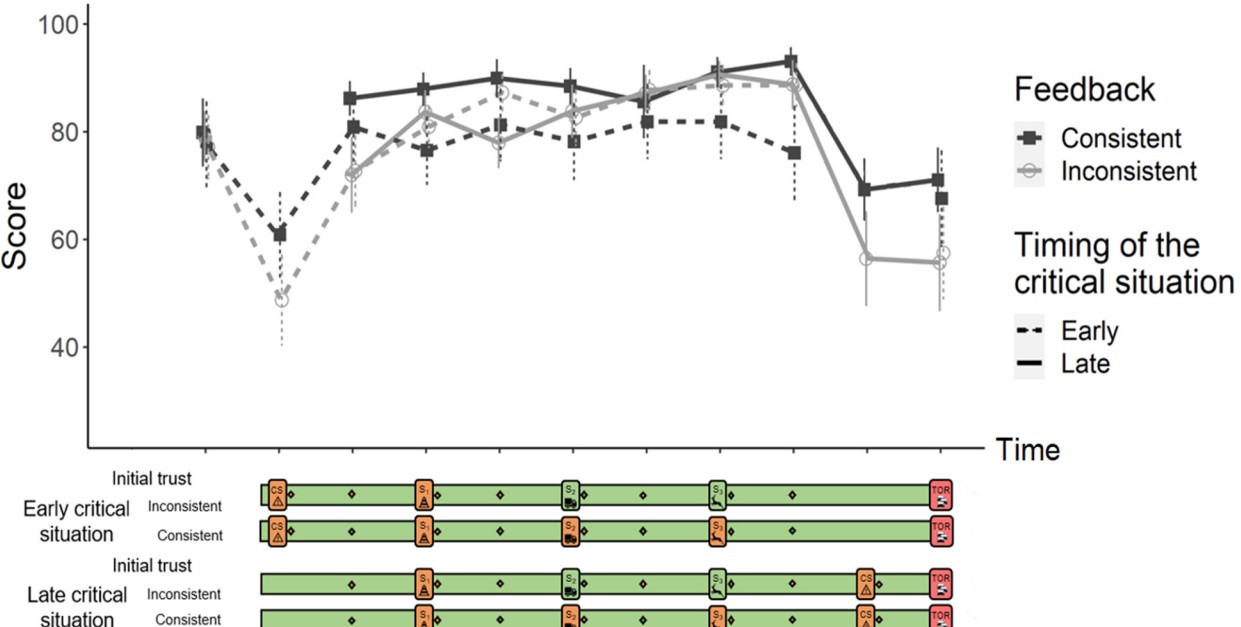

**Figure 4.** Evolution of drivers' level of trust during the drive, depending on the timing of the critical situation and the feedback consistency (error bars: standard error).

Finally, a repeated measure ANOVA was run with trust scores and perceived understanding scores to investigate whether the feedback consistency was correctly perceived by participants. It again exhibited a main effect of the Time, $F(4.98, 598.0) = 27.0$, $p < 0.001$, $\eta_p^2 = 0.184$, and an interaction between the Time and the type of measurement (i.e., trust vs. perceived understanding), $F(4.98, 598.0) = 3.70$, $p < 0.01$, $\eta_p^2 = 0.030$. This result indicates the two measures evolved differently over time. Trust increased during the drive while perceived understanding was lower in the case of inconsistent feedback, among drivers who were assigned to the inconsistent feedback condition (Figure 5).

Two linear models investigating, respectively, trust and perceived understanding were constructed to explore the effects of initial learned level of trust, age, gender, and Time (in minutes) during the drive. They also explored the several combinations of timing of the critical situation and feedback consistency. Moreover, the color of the feedback (green, orange, or red) at each measurement point was tested, alongside the timing of the critical situation.

The results (Table 4) indicated the initial learned level of trust had a positive influence on further trust evolution and perceived understanding of the HAD system. Each initial point of trust (on a scale from one to 100) had a strong probability ($p < 0.001$) to increase the further trust measurements of 0.46 points, and the other perceived understanding measurements of 0.40 points. Time also increased the trust measurements by 0.33 points for each passing minute but did not seem to influence perceived understanding. Age did

not seem to have any effect at all. Finally, all things being equal (i.e., neutralizing the effects of the initial learned level of trust, of the time, and of the age), males seemed to have a lower level of trust and lower self-assessed level of understanding of the automation than females.

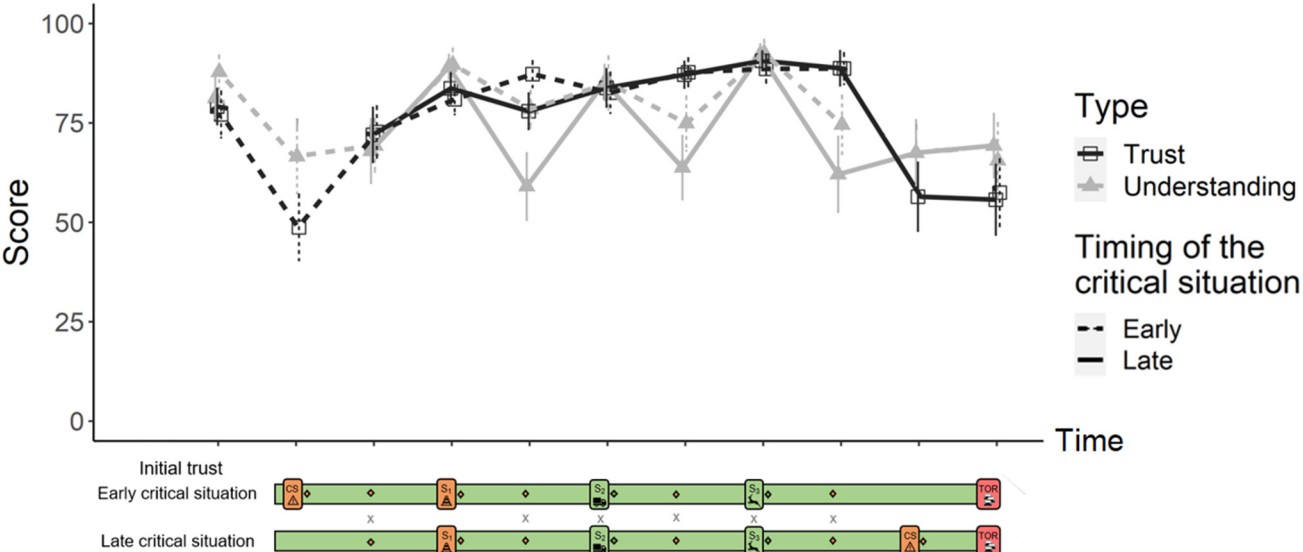

**Figure 5.** Trust and understanding scores for drivers who were assigned to the inconsistent feedback condition. Grey crosses indicate the moment of feedback inconsistencies (error bars: standard error).

**Table 4.** Linear models of trust and perceived understanding. ***: $p < 0.001$; **: $p < 0.01$; *: $p < 0.05$; ns: not significant.

|  | Trust | | Perceived Understanding | |
|---|---|---|---|---|
| (Intercept) | 40.50 | *** | 51.57 | *** |
| **Timing of Error and Feedback Consistency** | | | | |
| Early critical situation and consistent feedback | Ref. | | Ref. | |
| Early critical situation and inconsistent feedback | −4.50 | | ns | |
| Late critical situation and consistent feedback | 7.65 | * | ns | |
| Late critical situation and inconsistent feedback | −1.24 | | ns | |
| **Feedback Color and Timing of Error** | | | | |
| Green/HAD | Ref. | | Ref. | |
| Orange/HAD and early critical situation | 8.56 | | −16.47 | *** |
| Green/Correct situations and early critical situation | 9.23 | | −2.23 | |
| Orange/Correct situations and early critical situation | 1.39 | | −0.67 | |
| Orange/Critical situation and early critical situation | −17.33 | ** | −21.00 | *** |
| Red/TOR and early critical situation | −20.22 | *** | −18.43 | *** |
| Orange/HAD and late critical situation | 2.11 | | −24.92 | *** |
| Green/Correct situations and late critical situation | 7.05 | | 0.31 | |
| Orange/Correct situations and late critical situation | 2.41 | | 1.73 | |
| Orange/Critical situation and late critical situation | −25.87 | *** | −21.75 | *** |
| Red/TOR and late critical situation | −25.71 | *** | −20.88 | *** |
| Initial learned level of trust | 0.46 | *** | 0.40 | *** |
| Time (in minutes) | 0.33 | * | 0.10 | |
| Age | −0.00 | | 0.08 | |
| Male | −3.43 | * | −4.34 | * |
| Adjusted $R^2$ | | | 0.2998 | |

The results then confirmed the effect of the timing of the critical situation: drivers who had consistent feedback rated 7.65 points more at each trust measurement when they experienced a late critical situation, compared to the group that experienced an early critical situation. This result was not found for drivers who had inconsistent feedback.

Finally, the model indicated an effect of the feedback consistency and the color of the feedback on drivers' perceived understanding of the HAD. Inconsistent orange feedback during monotonous automated driving had a strong negative impact on perceived understanding, regardless of the timing of the critical situation. Drivers who experienced an early critical situation rated their perceived understanding 16.47 points lower than the reference situation (i.e., when the feedback was green and during monotonous automated driving). Drivers who experienced a late critical situation rated their perceived understanding 24.92 points lower than this same reference situation, suggesting that the last impression they had from the HAD was the most impactful. Nevertheless, no such effects were found on trust measurements. This result supports the idea that the mismatch between the HAD feedback and the situations was correctly perceived by drivers who experienced inconsistent feedback, but it had no main effect on their level of trust. Contrarily, the orange feedback during the critical situations and the red feedback during the TOR had a negative influence on both trust and perceived understanding, regardless of the timing of the critical situation.

### 3.2. Visual Behaviour

Drivers' visual behavior was then investigated during the several types of situations of the drive (i.e., HAD, correctly managed situations, critical situations, and TOR) and depending on the color of the feedback (Figure 6). Chi-squared tests were run to test whether the gaze frequency towards the several AOI were different during these combinations of colors and situations. The Chi-squared tests revealed that drivers' gaze frequency during HAD and when the feedback was green (Green/HAD) was different from all the other combinations of feedback color and situation ($p < 0.001$, Table 5). A difference was also observed when feedback was orange during correctly managed situations (Orange/Correct Situations) compared to the orange feedback while everything was normal ($p < 0.05$), the orange feedback during the critical situation ($p < 0.05$), and finally the red feedback during the TOR ($p < 0.05$, Table 5). These results confirm drivers correctly perceived feedback and monitored the driving environment accordingly. Drivers disengaged from on-board activities and increased their road monitoring when feedback indicated a potential problem.

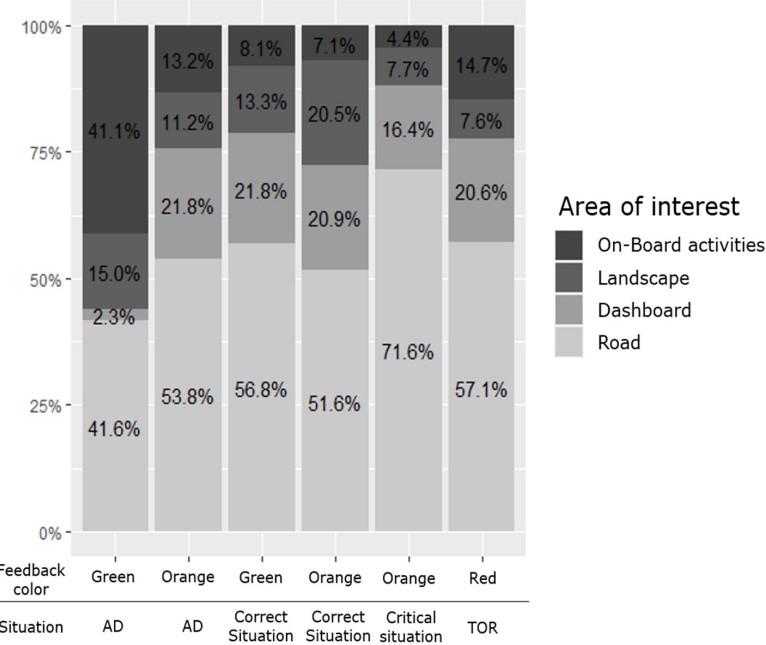

**Figure 6.** Proportion of drivers' gazes towards different area of interest. ***: $p < 0.001$; *: $p < 0.05$.

**Table 5.** Chi-squared results for the several comparisons.

| | Comparisons | $\chi^2$ | df | n | p | |
|---|---|---|---|---|---|---|
| Green/HAD | Orange/HAD | 32.23 | 3 | 61 | <0.001 | *** |
| | Green/Correct Situations | 40.36 | 3 | 61 | <0.001 | *** |
| | Orange/Correct Situations | 40.82 | 3 | 61 | <0.001 | *** |
| | Orange/Critical Situation | 50.53 | 3 | 61 | <0.001 | *** |
| | Red/TOR | 31.97 | 3 | 61 | <0.001 | *** |
| Orange/HAD | Green/Correct Situations | 1.48 | 3 | 61 | >0.1 | |
| | Orange/Correct Situations | 4.62 | 3 | 61 | >0.1 | |
| | Orange/Critical Situation | 8.34 | 3 | 61 | <0.05 | * |
| | Red/TOR | 0.902 | 3 | 61 | >0.1 | |
| Green/Correct Situations | Orange/Correct Situations | 1.87 | 3 | 61 | >0.1 | |
| | Orange/Critical Situation | 5.06 | 3 | 61 | >0.1 | |
| | Red/TOR | 3.5 | 3 | 61 | >0.1 | |
| Orange/Correct Situations | Orange/Critical Situation | 10.23 | 3 | 61 | <0.05 | * |
| | Red/TOR | 8.85 | 3 | 61 | <0.05 | * |
| Orange/Critical Situation | Red/TOR | 7.67 | 3 | 61 | >0.1 | |

## 4. Discussion

The present experiment aimed to investigate (a) the effect of the timing of a critical situation on trust construction, and (b) the effect of the feedback consistency on this trust evolution, depending on the drivers' initial learned level of trust.

### 4.1. Effect of the Timing of a Critical Situation

The timing of the critical situation had an influence on further trust evolution among drivers. Participants who experienced an early critical situation presented a trust increase at the next trust measurement, six minutes after that critical situation but no further increase after subsequent correctly managed situations. In comparison, drivers who experienced a late critical situation presented a trust progression during the previous correctly managed situations, confirming H1 and suggesting early critical situations may be a determinant in the drivers' further trust calibration process. This result is consistent with previous work that investigated other types of automations in the general TiA theory [15,16], and with previous results focused on TiAD [14]. Another experiment that showed a malfunction happening in the middle of the drive also led to a strong decrease in TiAD, which was recovered during the following minutes at the same level as before the malfunction [12]. In this last case, the malfunction did not prevent a further trust increase. It therefore seems malfunctions or critical situations decrease TiAD in the moment, but do not have a remaining negative effect when drivers were given the possibility to see the HAD system perform well before the malfunction. When drivers do not have the time to experience a properly functioning HAD system before the malfunction, TiAD progression seems affected for the rest of the experiment.

Otherwise, debriefings revealed most people did not understand why the ego vehicle zigzagged during the critical situation. When they were told about the missing lane marking, they were surprised to learn that HAD systems need this information to navigate properly. It therefore seems crucial for future study to inquire what drivers need to know about HAD functioning in order to ensure a proper fundamental knowledge of these technologies is spread in the drivers' population.

### 4.2. Effect of the Feedback Consistency

Previous studies showed that a HAD system displaying a level of uncertainty regarding its own capabilities may help drivers to calibrate their level of trust, particularly in situations where the HAD system is less reliable [18,19]. Moreover, other studies have suggested that using anthropomorphized feedback [22–24], or punctual messages explaining the reasons behind a specific HAD system behavior [25] may help drivers to calibrate their level of trust.

In the present study, drivers who experienced a consistent feedback had a stable trust during the drive. Comparatively, drivers who had an inconsistent feedback seemed to have seen a light increase in trust. Their first experience with the inconsistent feedback induced a lower level of trust that subsequently increased during the drive, although they indeed had trouble to understand this inconsistent feedback. This result invalidates H2 and suggests drivers' trust is not particularly sensitive to the feedback consistency, at least under the conditions of the reported experiment. This finding is at odds with existing literature on feedback importance [26,27]. Further investigations carried out on a larger sample of participants would be required to confirm this result. Such investigations would benefit from a detailed analysis of the tasks and visual behaviors engaged by people behind the wheel, including under real life automated driving conditions. In addition, debriefings showed that drivers frequently report they understood the HAD feedback (e.g., because there was a situation they did not perceive), even when there was nothing to understand. It is possible that drivers rationalize the HAD feedback a posteriori. In short, trust increases when drivers are given the opportunity to see the automation performs well, even when its features are not fully understood. An inconsistent feedback does not seem to have a negative influence on trust, and the performance of the automation appears to have the ultimate ability to influence trust. Future studies may also combine continuous and punctual feedback to fill the gap between the previous studies' results and the present ones.

Otherwise, a priori information related to an HAD system's reliability has been shown to influence the development of TiAD, either positively or negatively, before any interaction and during a video-based experimentation where participants experienced the same videos [28]. In the present study, participants had no particular a priori information, but experienced different types of feedback consistency that may have influenced the level of HAD reliability perceived by drivers. This consistency of the feedback did not seem to have a positive effect on TiAD, suggesting drivers' mental model and expectations may have a stronger influence on trust than actual HAD's feedback reliability. Of course, this conclusion is not irrevocable. Even if the colored lights used were able to attract visual attention efficiently and the color coding was very easy to interpret, a large variety of design options are conceivable and other designs may result in different effects.

Moreover, studies reported that missed detections from a lane departure warning system did not lead to significantly poorer manual driving performance [29,30]. In the present study, the vehicle's missed detections (i.e., green during correctly managed situations) are also likely to have a small effect on drivers' perceptions, because these situations were well-handled by the HAD system. On the other hand, a false positive (i.e., orange during monotonous automated driving) seems to disturb participants' TiAD at the first occurrence, but not after they saw the HAD system correctly detect the further situations on-road. This suggests that the drivers' "buffer" regarding false warning is quickly reset and does not load more discrepancies while the HAD system performs well. Additionally, previous studies showed these missed detections and false positives may have cumulative effects in case of several successive occurrences [29,30]. Future studies may explore if these cumulative effects also exist for TiAD. Nevertheless, it seems both ADAS and HAD do not need be entirely reliable for drivers to trust them.

### 4.3. Effect of the Initial Learned Level of Trust

The initial learned level of trust seemed to have an enduring effect on drivers' level of trust during their first experiences with the HAD system, confirming H3, as reported in

previous studies for drivers [13,14] and for passengers [31]. Nevertheless, no interaction between the initial learned level of trust and the time was found, contrasting with these previous results. The method that was used to obtain the two groups (i.e., Trustful and Distrustful) was different and may have led to smaller inter-individual differences between groups. In addition, the provided feedback may have had a stabilizing effect on drivers' TiAD, which would explain such results. Alternatively, the critical situation and the unexpected TOR may have participated to reduce the trust increase among distrustful drivers. Future studies are therefore needed in order to better understand the relation between the initial learned level of trust and other HAD-related specificities, such as the feedback consistency and the timing of critical situations, that may have interacted with it. Finally, no interactions between the initial learned level of trust and the timing of error, or the feedback consistency, were found, invalidating H4 and H5. This information may indicate the level of initial learned trust has a stronger effect than HAD's features and design (related to the dynamic learned trust, [8]) during drivers' first interaction with such automations.

### *4.4. Limitations and Perspectives*

The proposed experimental design regarding the early or late critical situation allowed us to investigate the trust development process during a 35-min drive. Nevertheless, the late critical situation was still relatively early, considering the driver's total experience time with a vehicle during its lifetime. Future studies may investigate TiAD construction and evolution over longer periods of time, with a first critical situation appearing after several positive interactions with an HAD system. Next, participants in the inconsistent feedback condition experienced several false positives between the situations that were correctly managed by the HAD system. Future studies may extend the present results by testing the dynamic evolution of TiAD when drivers are confronted with a succession of several false warnings without any other correct situations. Last, the current experiment used a median-split method to create the two groups based on the initial learned level of trust (i.e., Trustful and Distrustful). This approach provided a better vision of the general level of trust in the drivers' population compared to other grouping method based on pre-defined thresholds [13]). Nevertheless, the two groups were less representative of strongly trustful or strongly distrustful people, and the reported effects therefore must be carefully considered.

### 5. Conclusions

The present study suggests the provided automation's feedback has a much lower influence than the actual HAD performance, when considering drivers' trust formation process during HAD. Hoff and Bashir's [8] factor linked with the automation performance (i.e., the timing of malfunction) is more influential than the factor linked with the design features (i.e., the feedback consistency), confirming TiAD is mainly dependent of automation's ongoing performance. Nevertheless, it seems the initial learned trust is still preponderant during the first interaction with an HAD system, compared with factors related to the dynamic trust. Other studies investigating the trust formation and calibration process over longer durations (with, i.e., longitudinal studies on cohorts of drivers), could bring more information about the moment of transition when the dynamic learned trust gains more influence than the initial learned trust. Moreover, this information must be carefully considered when attempting to guide the drivers' trust calibration process. Modifying the vehicle's behavior in specific situations seems to be a more effective way to encourage drivers to recalibrate their trust than using feedback.

**Supplementary Materials:** The following supporting information can be downloaded at: https://www.mdpi.com/article/10.3390/info13100480/s1, Video S1: Critical situation (zigzag); Video S2: Critical situation (roadwork area).

**Author Contributions:** Conceptualization, J.B.M., M.B. and J.N.; methodology, J.B.M., M.B., J.N.; software, J.B.M.; validation, J.B.M., M.B., J.N. and R.B.; formal analysis, J.B.M. and R.B.; investigation, J.B.M.; resources, M.B. and J.N.; data curation, J.B.M.; writing—original draft preparation, J.B.M.; writing—review and editing, J.B.M., M.B. and J.N.; visualization, J.B.M. and R.B.; supervision, M.B. and J.N.; funding acquisition, M.B. and J.N. All authors have read and agreed to the published version of the manuscript.

**Funding:** This study was funded by the VEDECOM Institute.

**Institutional Review Board Statement:** The study was conducted in accordance with the Declaration of Helsinki and approved by the Ethics Committee of VEDECOM Institute (2020).

**Informed Consent Statement:** Informed consent was obtained from all subjects involved in the study.

**Data Availability Statement:** Not applicable.

**Conflicts of Interest:** The authors declare no conflict of interest.

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
