# Peer review of "Why Does the Automation Say One Thing but Does Something Else? Effect of the Feedback Consistency and the Timing of Error on Trust in Automated Driving"

_information, doi:10.3390/info13100480_

Round 1

Reviewer 1 Report

Information review 1830803

Many thanks for your contribution to an interesting and popular topic in transportation: trust in automated vehicles. This driving simulator study examined the level of trust and understanding of 61 individuals while driving an automated vehicle. Feedback on the status of the driving system was provided with an in-vehicle tablet and LED installed in the cockpit.  Gaze behaviour was also collected via glances in different areas of interest. The videos attached are helpful to understand what happened during the trials, this is a very good idea.

There are a few caveats to this paper that make me wonder whether it is suitable for publication. The paper does not provide a strong rationale for examining the research questions as they have already been explored, and some of these earlier works have been cited. Why would the authors pick one model of trust and not provide the reason supporting that choice? What about other models? In addition, the flow of ideas is not seamless, and the organisation lacks clarity, especially in the introduction. A few concepts are introduced (trust, transparency, timing) but the connections and transitions between them are missing. It is hard to tell what is novel about this research. The method section does not provide a fulsome description of what happened during the trial, and we don’t know whether the experimenters controlled whether participants looked at the driving environment or were engaged in a non-driving related task. This is barely mentioned in the gaze behaviour data at the end of the result section and comes out of the blue. That being said, the results are interesting and include self-reported data and more objective gaze behaviour data. However, the most intriguing results are not explained satisfactorily, and I have some reservations as regards the order of the variables used in the linear regression. Finally, the paper also needs proofreading. Additional comments are further detailed below:

·       L.15: Can a machine trust itself? I would refer to capability rather than trust.

·       L.35: What does a HAD system do?

·       L.37: Unnecessary self-citations for an argument that has been proven thousands of times, in many fields, and for decades.

·       L.40: Why did you choose the Hoff and Bashir model to serve as a basis for your study?

·       L.53: The authors do not explain why they focus on the concept of dynamic trust. Why is it more crucial than learned trust? They seem equally important and no rationale is provided to justify why this study focus on the dynamic aspects. The flow of ideas, in general, is hard to follow.

·       L.80: In terms of timing, level of trust, quantity?

·       L.81: What does "varied more dramatically" mean?

·       L.86: What is the novelty of this research as both the system status information and the timing of error effects on trust have been investigated in previous research?

·       L.87: What is a critical situation? What is an automation error? What is the difference between these two elements? Which one are you examining?

·       L.90: What are gazes used for? What are gazes measuring?

·       L.96: I guess a table would help clarifying the hypothesis.

·       L.101: Did you ensure none of them were colour-blind as your HMI relies on orange/blue/green lights?

·       L.106: Why would you ask about knowledge in ADAS system when participants experienced HAD?

·       L.114: The touchscreen is placed to the right of the steering wheel

·       L.131: Were participants asked to monitor the road? What were they allowed to do during the trial?

·       Fig.3: What does it mean when the diamonds change colours? Where participants given inconsistent feedback and asked to answer a question at the same time?

·       L.185: We need more details on this median-split groups: age, sex distribution, driving experience, HAD knowledge etc.

·       Table 1: Is distrustful the right term here? The 43.21 mean is very close to the median point of the scale, 50, which means these drivers do not distrust the automation, they are neither trustful nor distrustful. Furthermore, is there a conceptual reason that indicates that these groups, based on that criterion, are significantly different and should be compared?

·       L.213: Are you referring to "Trustful" drivers here? Please be consistent with the terminology

·       Figure 4: Please name the x axis

·       L.229: It does not increase linearly as shown in measures 4-5-6 late and 2-3-4 early

·       Table 2: Why is consistency in 5th and timing in 6th position when they are the main independent variables? Age and gender are exploratory variables have been inputted earlier in the regression analysis. It should be conducted depending on the order of importance/presentation of the concepts you examine.

·       L.261: Interesting result. It is peculiar that participants rated perceived understanding lower in the late condition than in the early. It may be that the last impression they had from the automated car was the most impactful.

·       Figure 6: It is surprising to see that participants were engaged in on-board activities. This should be introduced in the method section. If participants were engaged in on-board activities, how the experimenters ensured participants saw the HMI and paid attention to the vehicle's manoeuvres? This seriously questions the validity of the results.

·       L.292: Trust dropped after the critical situation, then it increased. That sentence is confusing

·       L.296: This is not about the first experience with an automated vehicles, this is about the timing of a critical situation.

·       L.306: Your results showed that trust scores bounced back after the critical situation, therefore affirming that "TiAD progression seems permanently affected" is not reflecting your observations and mislead the reader.

·       L.308: That may have had an effect on feedback consistency. Why is this not discussed?

·       L.321: What about participants' level of expertise in automated driving? This measure has been collected and could be added as a co-variate.

·       L.325: You do not provide an explanation for this result. Did all the participants saw all the events and the changes in the HMI? Was this controlled?

·       L.328: What does that sentence mean?

·       L.331: What about the realism of the driving simulation environment? It could also be a participant bias.

·       L.342: OK, but the nature of the feedback should also be discussed: why did you choose these colours and modalities? Could it be that a different feedback may have different effects on drivers? There is plethora of literature on this topic and this should be scrutinised.

Author Response

Dear Reviewer,

Please find our detailed answers in the attached PDF document.

Best regards,

The Authors

Reviewer 2 Report

Review Information-1830803, Why does the automation say one thing but do another?

Summary

The manuscript presents a study aiming to investigate how the timing of error and feedback affect trust in automated driving, using a driving simulator and a between-subjects design. Participants were split up in two groups using a median-split technique, having either low or high dispositional trust. Scenarios were presented with an automation lapse early or late in the scenario, and feedback was provided to participants repeatedly. The feedback was either consistent or inconsistent with the actual automation performance (both misses and false alarms were provided as feedback). Participants self-reported their trust in the automation and their understanding of the feedback. Also, participants’ gazing behaviour was recorded. It was found that automation performance had the strongest effect on trust in automation, and that inconsistent feedback did not affect trust in automation.

Comments

The topic is timely and interesting for the readership of Information. The manuscript is adequately structured and reads easily. The introduction is well informed, the hypotheses are adequately motivated, the methodology is mostly adequate, and the conclusions are adequately supported by the results.

One major concern I have is that initial level of trust is used as an independent variable. If I understand things correctly, participants filled a trust questionnaire at the start of the experiment providing information about their initial level of trust. Participants were then randomly assigned to the levels of the independent variables, Early vs late automation error and Consistent vs Inconsistent Feedback. A posteriori, participants were divided into two groups using a median-split technique, one group representing high initial level of trust and the other group representing low initial level of trust. Initial level of trust is then treated as the third independent variable, giving a 2x2x2 between-subjects design. With 61 participants, this gives roughly 6 participants per group, which is a rather small number for a between-subjects design. Furthermore, the authors provide no information about the number of participants per group, while, since a median-split technique is used, it cannot be assumed that the number of participants divides roughly equally over the eight different groups. Maybe the authors were lucky, but at least this information should be provided. Also, the mean (or median) scores for initial level of trust should be provided for the eight different groups, not just for the two trust groups as in Table 1, so that the reader can identify whether there are potentially relevant differences in dispositional trust between the different groups.

Treating the Dispositional trust variable as a dependent variable also allows conducting an analysis of variance. Since most of the scores are scale data, the authors should at least provide some evidence that assumptions for a parametric analysis are satisfied. Alternatively, it is not clear to me why dispositional trust is not treated as a Covariate in a Covariance analysis. This is not to say that one is necessarily better than the other in view of the questions that the authors want to answer. More simply, my point is that the authors should provide more argumentation about the choices that they made. 

Minor comments/language/grammar

A general point: I don’t know whether it is lack of grammatical proficiency or just sloppy editing, but the manuscript contains a rather high number of grammatical errors, and more careful proofreading before submission is encouraged (and appreciated by the reviewers!).

L11: role of human operator > role of the human operator

L24: drivers trust development > drivers’ trust development

L64: seem > seems (The grammatical subject is Feedback – although this is a case of ambiguity: maybe the authors’ intention is that the grammatical subject is Feedback … and malfunctions?)

L74 and L76: initial level of TiAD: I wasn’t sure whether this expression was meant to refer to dispositional trust or to initial learned trust. From reference [13] it appears that it refers to initial learned trust, but I would suggest using an expression that avoids confusion between the two forms of trust, and this consistent terminology should be applied throughout the manuscript.

L103: drove > drive

L104: drivers self-evaluated level > drivers’ self-evaluated level

L114: touchscreen was placed to the left: Figure 1 and the videos show that it was positioned to the right of the steering wheel

L132: Mancon et al, 2021 should be [13]

L140: latter > later

L190: somewhere else > somewhere else than

Figures 4 and 5: I suppose the error bars represent standard deviations. Please make this explicit in the captions

L243: each passing minutes > each passing minute

L251: each trust measurements > each trust measurement

L263: were > was (the grammatical subject is ‘mismatch’)

L263: drivers that > drivers who

L273: were > was (the grammatical subject is ‘gaze frequency’)

L301: same level than > same level as

L348; fist > first

L357: had > have

L365: futures studies > future studies

L370: have > has

L373: Limits > Limitations

L378: on > over

L392: have > has

L400: on > over

L400: with, i.e., cohort of drivers: unclear what is meant

L403: drivers trust calibration > drivers’ trust calibration

Author Response

(The authors gave the same response as above.)

Reviewer 3 Report

This manuscript describes a study to examine how automation feedback and failure primacy impacts driver trust calibration. Groups experienced a critical situation that was poorly handled by the automation either early (i.e., right after the beginning) or late (i.e., right before the end of it) in their trip.  They also examined feed-back consistency (consistent vs inconsistent).  They authors reported that early and poorly handled critical situation had an enduring negative effect on drivers trust development, compared to drivers who did not experience it.

The authors need to elaborate on why trust in automation is important.  They also need to explore the notions of trustworthiness, user confidence and appropriate trust in relation to safety. While developers want people to use and trust their products it is important to recognize and address the technical limitations of automation.  The focus here needs to be more user-centred.  Automation is intended to support people not vice-versa, which seems to be the approach here.  Users should not trust immature automation that is neither reliable nor predictable.

I struggled a little with the purpose of evaluating importance of feedback consistency (consistent vs inconsistent).  Empirical studies from over 30 years ago revealed that ineffective feedback can be a factor that contributes to poor calibration (e.g., Wagenaar and Keren, 1986; Norman, 1990; Cook et al., 1991). I think the conclusion that feedback was less important is misleading and very specific to system and situation.  Please discuss why it is important to investigate inconsistent feedback when it is already proven a bad thing.  

Minor English grammar correction throughout - the plural of feedback is feedback (no s).

Line 134 - Instructions to participants.  What instructions were given about interacting with the HAD?  Were there any non-driving tasks/ “on-board activities”?  It mentions a tablet, but I could not see a description of the NDRT if any.

Author Response

(The authors gave the same response as above.)

Round 2

Reviewer 1 Report

Thank you for addressing most of the comments. Despite significant improvements, I think the manuscript still needs to be English-proof as some errors remained.

Author Response

Thank you very much for all your work and feedback on this manuscript. We proof-read the manuscript again, chasing grammar mistakes.

Reviewer 2 Report

I am happy to see that the authors have addressed my comments to the initial submission, and I have no remaining major comments. However, there are a few additional points concerning the presentation (to be elaborated below), that require attention before the manuscript can be published. 

Minor comments

L110ff: The motivation for the hypotheses concerning gaze is missing. Also, it appears that H4 and H5 are dependent on H3. I would prefer to have hypotheses that are formally expressed as statements that can be accepted or rejected as such.

L158: Given that the authors have decided that the kind of trust they focus on is initial learned trust, it would be good to have the items of the trust scale by Manchon et al included in a table, allowing the readers to get an impression of what initial learned trust is about

L373-375: When drivers do not have …, TiAD progression seems affected on the long term: On the basis of the presented results, this is strongly overstated

Presentation

The Results and Discussion sections are somewhat lengthy, and therewith momentum is lost. In my opinion, the parts (including the associated tables) about the effect of the different colours may be left out. In that way, momentum would be maintained and the main message, about the effects of automation performance and feedback, would come out more clearly.

As before, there are quite a few grammatical errors. Interestingly, they are not evenly spread through the manuscript. This suggests that different authors have contributed different parts and that the proficiency in writing English is not the same for the different authors. In such a case, I would have hoped that the authors would have checked each other’s contribution for grammar  … Finally, it is okay to have a version with all changes marked, but since the cumulated result of deletions and insertions at times is rather obscure and confusing, it would be nice having a clean version as well. 

Language/grammar

L22: sometimes-indicated > sometimes indicated

L37: a level 4 > level 4

L37: a fully manual task, to > a fully manual task to

L43: that … that … : unclear relation of restrictive relative clauses: does the second that-clause (“that directly influences reliance on automation”) relate to “psychological construct” or to “human attitude towards automation”?

L44: influence > influences

L46: to select one factors-centered model which structure: unclear. Maybe: to select a factors-centered model , the structure of which. Or leave ‘structure’ out altogether, because it is sufficient to say that the model served as a basis for research

L52: from a situation to another > from one situation to another

L53: knowledges > knowledge (‘knowledge’ is a non-countable noun)

L56: refers to: rather vague. Maybe: ‘is influenced by’ or ‘depends on’?

L64: require > requires

L77: a > an

L77: influent > influential

L78: have > has

L84: allow > allows

L100: homogenous > homogeneous

L114: are > is

L133: the > they

L176: making > marking

L180: the obstacle: an obstacle was not mentioned before. Therefore, a definite determiner is not correct

L242: show > shows

L263, L267, L268, L277, L279: When reporting F values, it is common practice to first mention the degrees of freedom for the treatment and then the degrees of freedom for the error term, so F(1,59) instead of F(59,1)

L312, L319, L322, L327: drivers that > drivers who

L359: participants that > participants who

L360: presented a trust increase sight after that critical situation: not fully correct and misleading. The measurement showing the increase was 6 minutes after the critical situation

L363: “suggesting HAD system critical situation": unclear what is meant

L392: furtherly: subsequently

L451: have > has

L454: Limits > Limitations

L475: influent > influential

Author Response

Please, find our detailed answer on the attached file.

Reviewer 3 Report

The authors have adequately addressed my concerns in their revised manuscript. 

Author Response

Thank you very much for all your work and feedback on this manuscript. We are glad the previous revision answered your questions.